# Geomorphosites as Geotouristic Resources: Assessment of Geomorphological Heritage for Local Development in the Río Lobos Natural Park

Rosa María Ruiz-Pedrosa [1,*], María José González-Amuchástegui [2] and Enrique Serrano [1]

1   Geography Department, University of Valladolid, 47011 Valladolid, Spain; e.serrano@uva.es
2   Geography and History Faculty, Distance Education National University (UNED), 28040 Madrid, Spain; mjgonzalezamu@geo.uned.es
*   Correspondence: rosamaria.ruiz.pedrosa@uva.es

**Abstract:** Natural protected areas (NPAs) are territorial resources that have received an increasing number of visitors in societies with a high demand for landscapes of high aesthetic and scenic value. Tourism is one of the main activities in NPAs, and within this, geotourism plays an important role, becoming an effective resource in the promotion of natural heritage with repercussions on local and regional economic development. The aim of this work is to analyse geomorphosites' tourist potential in natural protected areas, focusing on the case of the Río Lobos Natural Park (Castilla-León, Spain) and its geotourism cartography, as well as the proposal of different geotourism routes. To this end, a methodology is applied to the 14 geomorphosites inventoried in the Cañón del Río Lobos, based on a combination of different methods. Its application results in a classification with three thresholds (high, medium and low) for each geomorphosite analysed. In addition, a series of management proposals are included. The validity of this methodology applied for the evaluation of the tourist potential of geomorphosites endorses its application for other natural protected areas.

**Keywords:** natural protected areas; geomorphosites; geotourism; local development; geotourist map

## 1. Introduction: Natural Protected Areas, Geomorphosites and Tourism

Since the mid-2000s, interest in the value of geomorphological heritage and its relationship to geotourism and conservation has grown worldwide. This can be seen in the Geomorphosites and Landform Assessment for Geodiversity working groups of the International Association of Geomorphologists, which have mainly focused on the development of this specific field of research on geomorphological heritage and have promoted geomorphosites as key sites for education and tourism.

Geomorphosites are geomorphological heritage sites that are of particular importance for the understanding of Earth's history, are spatially unlimited and are clearly distinguishable from their surroundings [1]. The importance of geomorphology as heritage is derived from its status as infrastructure for habitats and landscapes, as it supports lifestyles and cultural elements and has continuity on Earth's surface [2].

Geomorphological heritage is often framed in terms of natural protected areas (NPAs). The first NPAs arose from the idea that our natural heritage should be unlimited for all to admire and enjoy, protected and inherited from generation to generation. Thus, we have come to recognize the existence of exceptional spaces that stand out for their beauty and value, whose preservation is recognized as a common objective of society. In NPAs, geomorphological elements have territorial and landscape components that differentiate them from geological sites. Geomorphosites attract travellers, hikers and tourists who wish to understand the territory they are visiting, and they are attractive territorial resources. Local societies are also attracted to landforms that are present in their lives, but often these are not rightfully recognized in terms of their environmental and cultural relationships.

NPAs landscapes are defined by their relief and landforms, the condition of their ecosystems or habitats and the organization of their territory [2–5].

Geomorphosites are related to tourism due to their high aesthetic and landscape value at all scales, with many geomorphosites in the world attracting thousands and millions of tourists. Their popularity is always linked to their exceptional aesthetic values or very active dynamics (volcanoes, glaciers or canyons) that encompass the beauty of nature. When speaking about NPAs, we, therefore, are often referring to nature tourism, in which natural elements and landscapes play leading roles and are closely linked to development and sustainability, as well as the involvement of the local population. Nature tourism encompasses different tourism models, such as rural tourism, sustainable tourism, ecotourism and active tourism. Therefore, geomorphosites are integrative elements essential to our understanding of a territory, its complex relationships with human activity (either limiting or favouring it), human and artistic development and history on a local scale [2,5].

The concept of geotourism is more recent, although in the last decade, it has experienced growth and has been addressed by numerous authors worldwide [6–10]. Geotourism was defined in the Arouca Declaration as "tourism that sustains and enhances the identity of a territory, taking into account its geology, environment, culture, aesthetic values, heritage and well-being of its residents" (Arouca Declaration, 2011). Geotourism is appreciated and accepted as a useful tool to promote natural and cultural heritage, in addition to fostering local and regional economic development, especially in rural areas [11]. The demand for tourism products linked to geoheritage exploitation in NPAs and interest in the study of the geotouristic potential of geomorphosites are increasing [6,7,12–15]. A wide range of assessment methodologies to determine the geotourist potential of geomorphosites and to improve their management and conservation have been developed over the past few years [6,11–13,16–21].

The most important geomorphological elements of NPAs must be determined from scientific and managerial points of view, as well as their contribution to the internal management of protected areas and local environmental development. Specific tools are required, including maps, geotourism and interpretation proposals and routes [22].

The objectives of this work are (1) to inventory the geomorphosites of the Cañón del Río Lobos Natural Park; (2) to evaluate their potential as a tourist resource and reflect this in a geotourism map; and (3) to design and propose georoutes whose objective is to promote the geomorphosites of the park.

## 2. Methodology: Geomorphosites' Tourism Potential Assessment, Tourist Map and Geotourism Routes

### 2.1. Geomorphosite Inventory

Our study of geomorphosites in the Río Lobos Natural Park starts by outlining the geomorphology of the study area, applying techniques used in geomorphology and showing its geomorphological cartography in detail at the 1:10,000 and 1:25,000 scales (for more information, see [2]). This is not only a geomorphological study of the park but also a consideration of its cultural heritage and its territorial implications, considering additional aspects of use and management, as well as the value of its geomorphosites as a tourist resource.

An assessment of its geomorphological heritage is carried out following a methodology that has already been applied for mountain and rural landscapes [2,3,14,22,23]. It is based on the inventory of geomorphosites and an assessment of their intrinsic or scientific value, added value, and use and management values.

### 2.2. Geomorphosites Tourism Potential Assessment

After selecting and inventorying geomorphosites, they were evaluated as a tourist resource, focusing on their potential for enjoyment, leisure and learning about geomorphology. The assessment prioritized the practical use and management of the sites over their

intrinsic values [12]. This led to the development of a geotourist map as a practical tool (Section 4.3: Geotourism map and routes).

The assessment considered a combination of values identified by previous authors [6,12,15], including scenic, scientific, cultural, educational, conservation (vulnerability, use limitations) and added value. This comprehensive approach aimed to provide a broad understanding of the natural, cultural and practical aspects of the geomorphosites (Table 1).

**Table 1.** Methods for tourism potential assessment review.

| Values | | Authors | | | |
|---|---|---|---|---|---|
| | | **Pralong [12]** | **Kubalíková [6]** | **Brilha [15]** | **This Work** |
| Scenic | | Yes | No | Yes | Yes |
| Scientific | | Yes | Yes | No | Yes |
| Cultural | | Yes | No | No | Yes |
| Economic | | Yes | Yes | Yes | No |
| Educational | | No | Yes | No | Yes |
| Conservation | Vulnerability | No | Yes | Yes | Yes |
| | Use limitations | No | Yes | Yes | Yes |
| | Singularity | No | Yes | Yes | No |
| Added | Accessibility | No | Yes | Yes | Yes |
| | Security | No | Yes | Yes | Yes |
| | Logistics | No | Yes | Yes | No |
| | Population | No | Yes | Yes | No |
| | Observation conditions | No | Yes | Yes | Yes |
| | Proximity to recreational areas | No | Yes | Yes | Yes |

Following Brilha [15], a numerical assessment proposal was assigned to each value, giving more importance to values appreciated by tourism, such as scenic, cultural, conservation, accessibility and safety (Table 2). Cultural values are considered in relation to existing cultural assets, which complement or enhance geomorphosite assessment because of the mutual interrelationships between geomorphology and historical–artistic or cultural elements.

**Table 2.** Tourism potential assessment and assigned values.

| Values | | Importance | Numerical Assessment Proposal |
|---|---|---|---|
| Scenic: panoramic view, size of panoramic view, geographic diversity, natural diversity | | 15 | 0–5–10–15 |
| Scientific: integrity, rarity, geodiversity, scientific knowledge | | 10 | 0–5–10 |
| Cultural: presence of cultural values, value of cultural elements, number of elements, historical diversity | | 15 | 0–5–10–15 |
| Educational: representativeness and clarity of forms or processes, pedagogical exemplarity, didactic documentation available, current educational use | | 5 | 0–5 |
| Conservation | Vulnerability: risk of degradation and fragility | 10 | 0–5–10 |
| | Limitations on use: legislation for its protection | 5 | 0–5 |
| Added | Accessibility | 10 | 0–5–10 |
| | Security | 10 | 0–5–10 |
| | Observation conditions | 15 | 0–5–10–15 |
| | Proximity to recreational areas | 5 | 0–5 |

*2.3. Geotourism Map Design*

After assessing geomorphosites' tourism potential, a geotouristic map was created by simplifying the geomorphological map and adding tourist information, taking into account the criteria of target audience, objectives, graphic choice and practical aspects of use that favour its usefulness and applicability [24]. Among the various types of maps and proposals for geotourism, geodidactic maps are the most suitable for education and recreation. These maps, which are large-scale, can be reproduced in brochures and leaflets, with schematic backgrounds, figurative symbols and additional information relevant to tourists [25,26]. The maps are focused on tourists not initiated in geomorphology or Earth sciences, facilitating the understanding of the geomorphological or geological phenomena visible in geomorphosites.

Geotouristic maps are designed to highlight recognizable landscape features and be simple, clear and practical for use in the field [27–30]. Both principles are synthesized to provide visitors with a useful map for discovering and understanding abiotic elements, while maintaining scientific rigor, as a document for dissemination and scientific knowledge. The preparation of detailed geotouristic maps with interpretive routes or trails should prioritize geoconservation as a key objective. This involves selecting key elements of the geomorphological map, simplifying reading levels and spatially representing the most significant elements [24,26–28,30].

## 3. Study Area: Río Lobos Natural Park

*3.1. Río Lobos Natural Park*

The Río Lobos Natural Park (RLNP), declared in 1985, is located between the provinces of Soria and Burgos, nestled between the southern foothills of the Iberian Mountain Range, in the transition between the Duero plain to the south and the Urbión and Neila mountains to the north (3°6′40″ W/41°47′7″ N, Figure 1). Covering an area of approximately 12,244 hectares, the justification for its declaration and protection lies in a spectacular karst landscape in a mid-mountain environment, characterized by a 26 km long calcareous canyon with steep walls. The traditional activities of the resin industry and cultural elements have left a human footprint on the landscape, and in recent years, the area has been used for forestry and geotourism.

It is one of the first NPAs declared in Castilla y León county. The Canyon had previously been included in the National Inventory of Outstanding Landscapes (1975) and in the Open Inventory of Natural Areas of Special Protection (1980). The 1985 Decree of Declaration of the Park established the delimitation of the park, within which two zones of the Natura 2000 European Network are recognized: the Site of Community Importance "Cañón del Río Lobos" (European Habitats Directive) and the Special Conservation Area for Birds "Cañón del Río Lobos" (European Birds Directive).

The RLNP is located in two distinct counties: Pinares and Burgo de Osma. These municipalities are home to a total of 3187 inhabitants (as of 1 January 2022), with the most populated being San Leonardo de Yagüe, where 64% of the park's inhabitants live. The socioeconomic area of the park is characterized by a clearly aging and regressive population, with the bulk of the population between 35 and 59 years of age, very low birth rates and a higher proportion of women than men, especially after the age of 65, due to the greater longevity of women [2].

The economy of this area is primarily based on the exploitation and transformation of wood. The primary sector focuses on forestry, particularly wood exploitation, and, to a lesser extent, on crops and livestock. In recent years, mycology has also become increasingly important. In the tertiary sector, geotourism or natural tourism and related activities and services are prominent.

The prevalence of limestone and karst processes explains most of the karst landforms in the park, both on the surface (exokarst), such as sinkholes and karrens, and below the surface (endokarst), such as caves and chasms. The karst environment dominates the landscape and determines the distribution of ecosystems and human uses that are

adapted to the geomorphological conditions. The area features a complex structural relief defined by the presence of perched synclines, synclinal platforms and thrust faults. These landforms have been modified by tertiary erosion surfaces, on which karstic processes have developed.

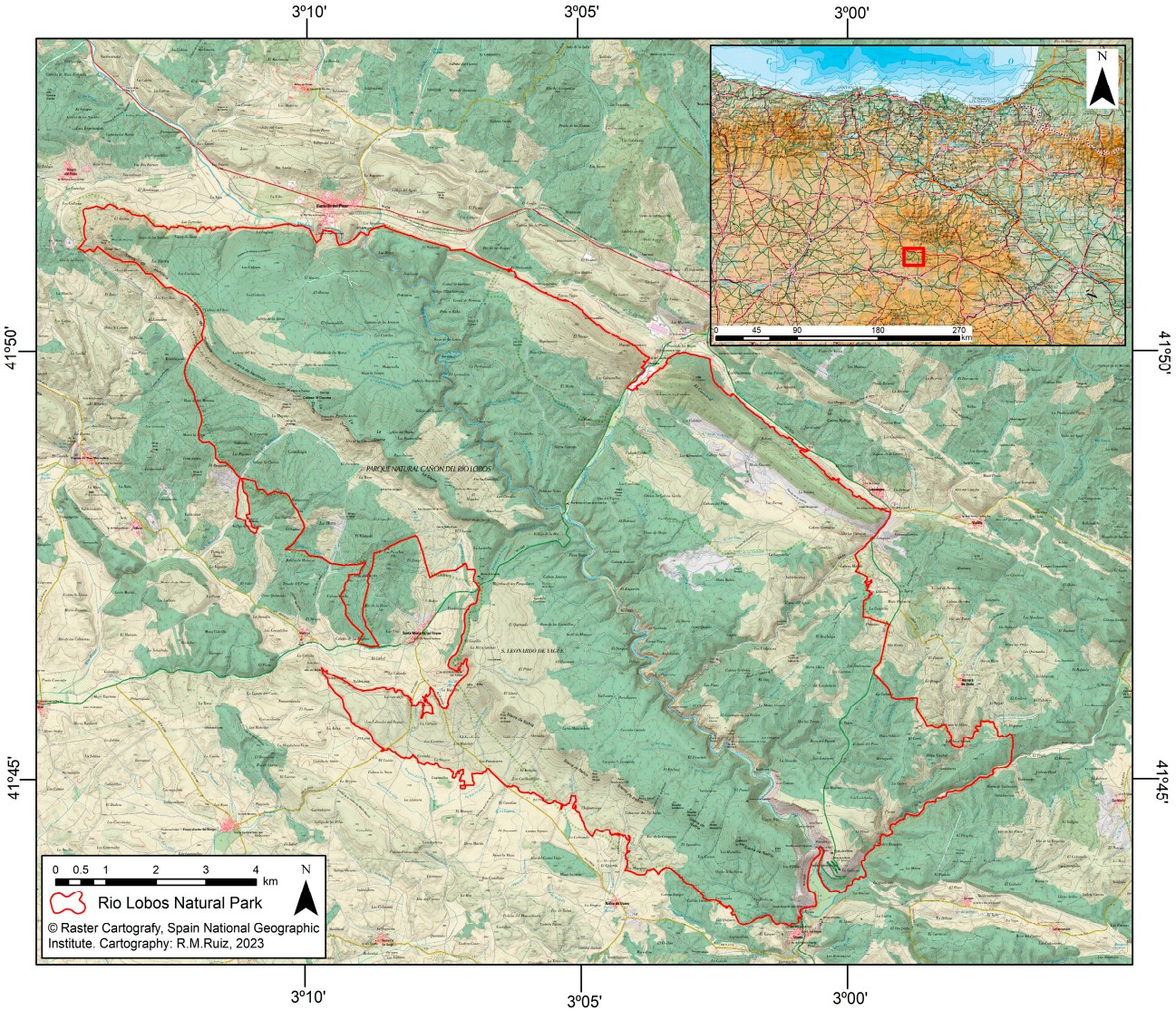

**Figure 1.** Río Lobos Natural Park boundaries, located in Soria Province, Castilla y León region, Spain.

The RLNP also contains cultural elements that enhance its landscape value, including five Sites of Cultural Interest recognized by the Castilla y León regional Government, as well as inventoried archaeological sites. San Bartolomé hermitage is particularly noteworthy, with its outstanding monumental character due to its artistic and historical value and its location at the entrance of the canyon. It has been recognised as a Historic Artistic Monument since 1983 and as an Asset of Cultural Interest with the category of Monument in 2015. This Romanesque construction from the 13th century is characterised by its Templar symbolism and esoteric figures.

Overall, RLNP is a rich and diverse space where karst geomorphology is the protagonist of the landscape, where natural and cultural heritage coexist, creating sites of extraordinary cultural, natural and scenic value, with high aesthetic appeal for visitors.

### 3.2. Tourism in Río Lobos Natural Park

Since its declaration, RLNP has become a popular tourist destination, despite not being a geopark or a tourist spot. As a NPA, its primary goals are conservation, education, public enjoyment and improving the quality of life of its inhabitants. In 2006, it was recognized by the Spanish Tourism Quality Institute for its services, facilities for public use and management policies aligned with conservation objectives.

Massive visits began in the 1980s, coinciding with its declaration, and since then, the interests of visitors have been diverse. Tourists are not a homogeneous group in any sense, neither economically, socially nor culturally. Tourism in the RLNP can be categorised into three groups: mass tourism, cultural tourism and active tourism. Mass tourism is the most prevalent in the park, concentrated in recreational areas and in the vicinity of Ucero, with a focus on enjoyment, recreation and socialization, being short visits that massify very specific places in the park, such as San Bartolomé hermitage. Cultural tourists are interested in the park's cultural and natural heritage; they are much more aware of conservation and sustainable use and visit multiple locations to learn about RLNP values. They consume the park's interpretive and informative products. Active tourists seek physical activity and sports satisfaction and are less interested in interpretation and knowledge-based experiences (hikers, runners, cyclists, climbers. . .) [2].

The RLNP keeps two different records of the number of visitors: on one hand, visitors who come to the Interpretation Center (Figure 2), and on the other, through four pedestrian gauges and one vehicle gauge distributed throughout the park, one of which is located at the San Bartolomé hermitage. The Interpretation Center was the most visited in the community in 2021, with 38,801 visitors, despite the restrictions and limitations caused by COVID, which forced its closure in the high season of spring and prevented the visitation of tourists from other autonomous communities until the end of May. In 2022, visitors fell to 32,748, still making it the second most visited Natural Interpretation Center in Castilla y León. Therefore, it has been able to recover the flow of visitors prior to COVID, which in 2019 was 38,007.

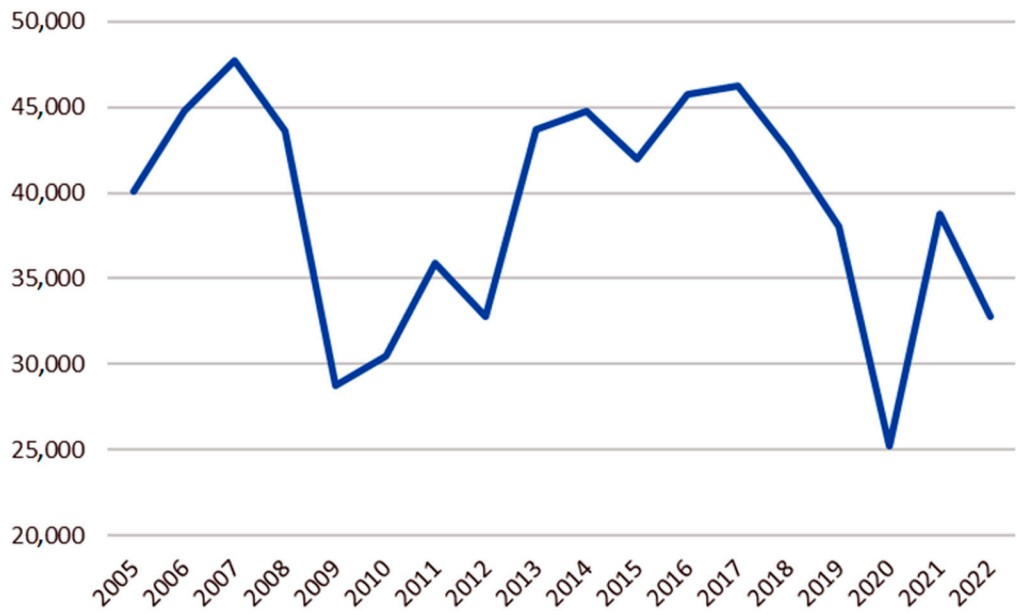

**Figure 2.** Visitors of the RLNP Interpretation Center between 2005 and 2022.

The data from the Interpretation Center at RLNP are affected by the visitors count from the gauges located in the park, which recorded a total of 151,015 visitors in 2022, a significant decrease from 313,966 in 2019. In 2019, only 12% of visitors visited the Interpretation Center, while in 2022, this percentage rose to 21%. In summary, RLNP has lost half of its pre-pandemic visitors, but visitors are more interested in interpretation

and seek information and guidance. This indicates a shift in the tourism model, from independent tourists to those seeking knowledge.

The majority of tourists are families and social groups, engaging in contemplative and leisure nature tourism, with a growing focus on interpretation and learning. School groups and cultural associations, previously made up around 10% of visitors, accounted for only 2% in 2021, indicating a decline in organized educational visits due to the pandemic. Additionally, there are visitors attracted to active and adventure tourism activities, such as horseback riding, cycling and caving, focused on a few caves in the park. In 2021, 25 authorizations were granted for caving activities, with the tourist cave of La Galiana being the main attraction for organized active tourism, operated by a concession to a private company.

Visitors flow is spread out across all months of the year, with peaks in August (18% of visitors in 2022), Easter (15%) and certain long weekends. The highest influx of visitors coincides with long weekends and vacation periods, with weekends accounting for 52.3% of annual visits, leading to crowded conditions on certain dates, indicative of mass tourism.

In 2022, the vast majority of visitors to the Interpretation Center were of national origin (over 98%), with Madrid being the primary source of visitors (36%), followed by Castilla y León (21%), Valencia (9%) and Catalonia (7%). Of the 2% of foreign visitors, 33% came from France, followed by the United Kingdom, Germany and the Netherlands, each accounting for 11%.

## 4. Results: Geomorphosites as a Geotouristic Resource in Río Lobos Natural Park, Spain

### *4.1. Geomorphosite Inventory*

Following the methodology outlined in Section 3, a total of fourteen sites have been inventoried in the RLNP. These sites have been identified as potential geomorphosites based on their geomorphological and landscape features. A description sheet has been created for each site, detailing its intrinsic values (scientific content), natural dynamics, uses, impacts on the geomorphosite and added values (cultural, educational, touristic, use and management). The description sheet also includes location information and images [2].

Geomorphosites are classified based on their scale, categorized as "Elements" when a landform is of interest by itself or "Places" when it is an association of landforms of different ages and genesis. Additionally, they are categorized as "Exceptional" if they are an exception in the RLNP as a whole and are very poorly represented or "Representatives" if they symbolize the overall characteristics of the territory and provide a general understanding of the geographical, geomorphological and landscape features of the NPA.

Each geomorphosite has also been attributed a geomorphological classification as well as accessibility and interest values (Table 3). The predominant geomorphological attribution is structural (five cases) and karstic (five cases), both classifications representing the 71% of the total, due to the presence of folded and calcareous materials. Fluvial geomorphosites and an example of interesting slope processes are less common.

**Table 3.** Geomorphosites inventoried in Río Lobos Natural Park.

| No | Geomorphosite | Geomorphic Attribution | Type | Character | Accessibility | Interest |
|----|---------------|------------------------|------|-----------|---------------|----------|
| 1 | Río Lobos–San Bartolomé Canyon | Fluviokarstic | Place | Representative | High | High |
| 2 | Arganza fault-line valley | Fluviostructural | Place | Exceptional | High | Medium |
| 3 | La Sierra syncline flank crest | Structural | Place | Representative | Low | High |
| 4 | Virgen de la Cueva syncline flank | Structural | Place | Representative | High | High |
| 5 | Pico Navas slopeslide | Slopes | Element | Exceptional | Medium | High |
| 6 | Pico Navas perched syncline | Structural | Place | Representative | Medium | High |
| 7 | Las Raideras river sink | Karstic | Element | Representative | High | High |
| 8 | Hoyo de los Lobos syncline flank | Structural | Place | Exceptional | Medium | Medium |
| 9 | La Isla entrenched meander | Fluvial | Place | Representative | Medium | High |

**Table 3.** *Cont.*

| No | Geomorphosite | Geomorphic Attribution | Type | Character | Accessibility | Interest |
|---|---|---|---|---|---|---|
| 10 | Costalago orthocline valley | Structural | Place | Exceptional | High | High |
| 11 | Las Tainas y el Torcajón pits and karstic area | Karstic | Place | Representative | Medium | Medium |
| 12 | La Galiana karstic system | Karstic | Element | Representative | High | High |
| 13 | Ucero springs | Karstic | Element | Representative | High | Medium |
| 14 | Chorrón fluvial sink | Karstic | Element | Representative | Medium | High |

The five geomorphosites with structural attribution are representative of folded relief, including perched synclinal and syncline flank crests (Figure 3). N° 3, the La Sierra synclinal flank crest, and N° 4, the Virgen de la Cueva synclinal flank, represent the SW and N flank ridges, respectively, of the Río Lobos syncline. The perched syncline of Pico Navas, geomorphosite N° 6, raised to the NW, adds to the structural features, the presence of karst landforms and processes, dominated by karrens, slope dynamics and the added cultural interest derived from the remains of the wall of the Celtiberian fortified settlement. The Costalago orthocline valley geomorphosite (N° 10) has been qualified as singular for its great structural, landscape, lacustrine and fluvial interest, with active dynamics of incision, solifluxion and landslides.

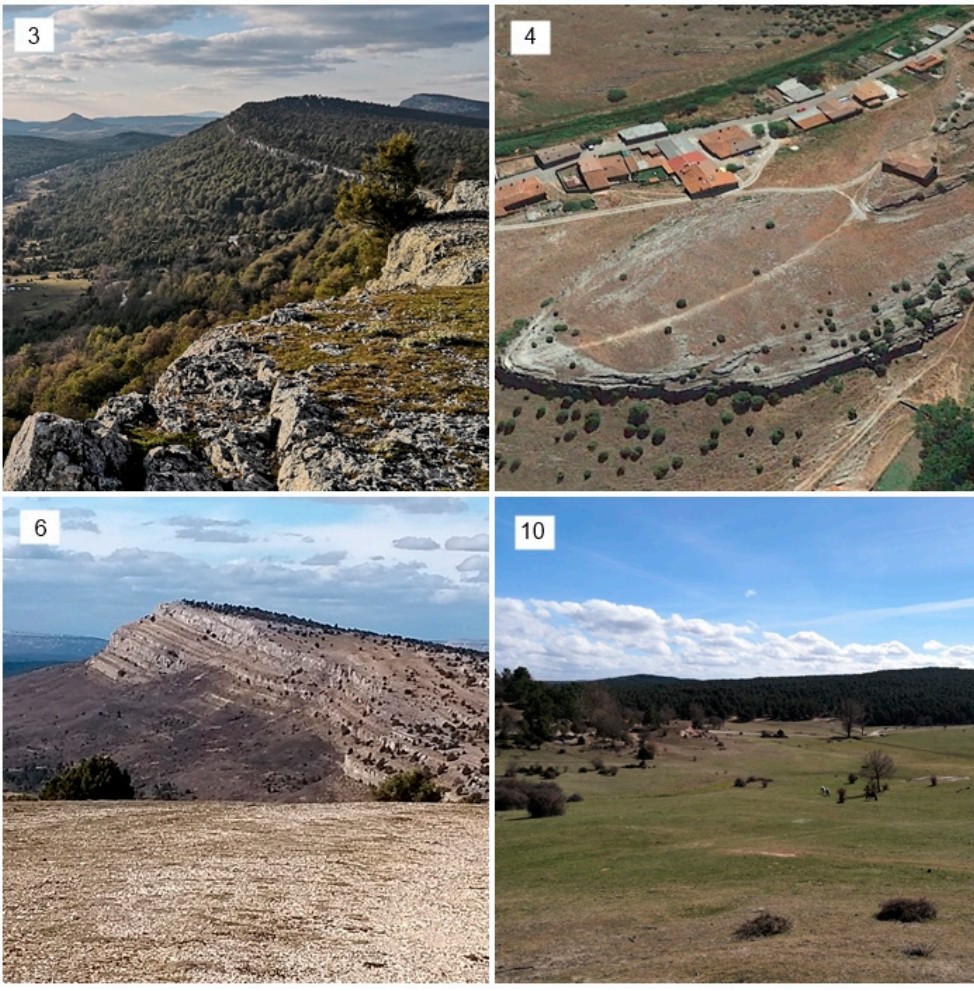

**Figure 3.** Structural attribution geomorphosites (N° 3, 4, 6 and 10).

The karst or fluviokarst geomorphosites are the most significant landscapes in the park, featuring karst canyons, sinkholes, chasms, caves and springs (Figure 4). The most

well-known and representative geomorphosite is the Río Lobos–San Bartolomé Canyon (N° 1), a system formed by a holokarstic canyon with vertical walls and traces of karstification levels, horizontal hanging caves such as the Cueva Grande and detrital and calcareous fillings in its interior. It is also of great cultural interest due to the presence of the San Bartolome hermitage and cave paintings.

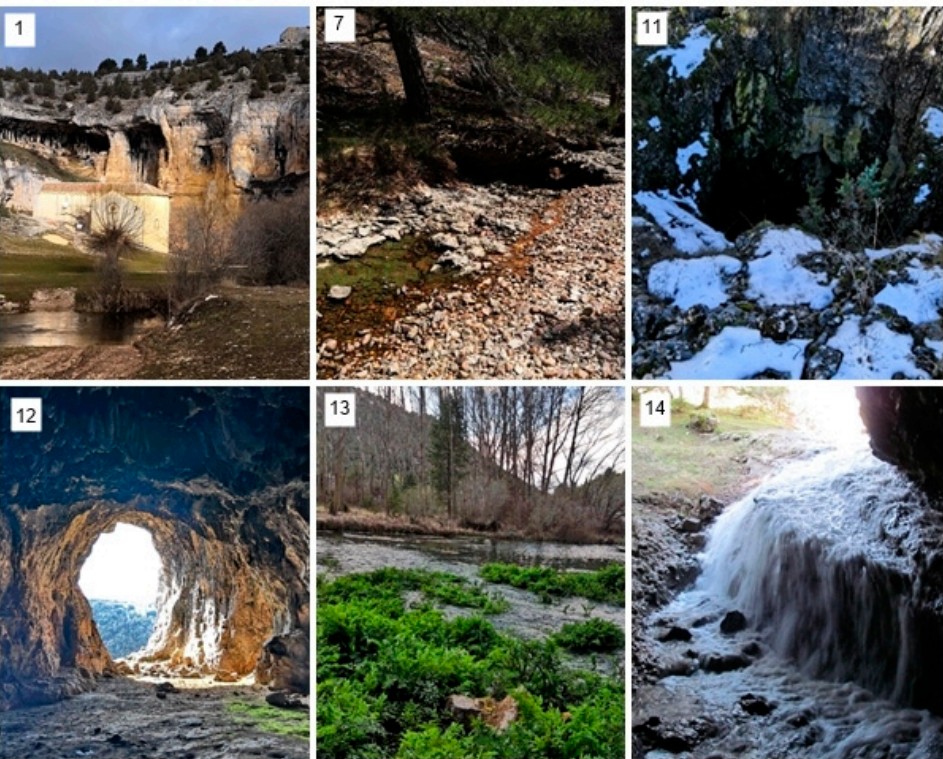

**Figure 4.** Karstic and fluviokarstic attribution geomorphosites (N° 1, 7, 11, 12, 13, 14).

Two geomorphosites are characterized as being karst sinks, which are significant elements of the park's geomorphology and hydrology. These are N° 7, the Las Raideras river sink, and the Chorrón fluvial sink, N.°14. Las Raideras is a karst sink with progressive seepage into fluvial deposits and limestones, with flow losses of up to 400 L/s [31]. It is responsible for the absence of river flow in the Lobos River up to downstream of the Siete Ojos Bridge. El Chorrón is a sink where the water sinks into the karstic system of the Lobos River, forming a horizontal mouth through which the water flow enters.

Geomorphosite N° 11 represents Las Tainas and el Torcajon pits and karstic areas, where exokarstic elements (structural karren, sinholes and captured karst depressions) and endokarstic elements (vertically developed chasms, such as the Tainas and Torcajón, 100 m deep) adapted to the structural conditions are developed. Similarly, the geomorphosite N° 12 of La Galiana has an endokarst system characterized by the existence of cavities with predominantly horizontal development. The Ucero river spring (geomorphosite N° 13) has a karstic–fluvial attribution, as it is a karstic upwelling where most of the waters of the Lobos River karstic system outcrop, feeding the Ucero river.

In addition to the fluviokarst geomorphosites, there are also sites that exemplify the sinuous and meandering hydrographic network typical of the park. These are the geomorphosites of fluvial attribution (Figure 5). Notably, site N° 9, the La Isla entrenched meander, is an example of a highly sinuous entrenched meander, with steep walls, a flat bottom occupied by fluvial terraces and a dry riverbed for most of the year. There are two geomorphosites with both fluvial and structural attribution, where tectonics has directed the fluvial network. These are sites N° 2 and N° 8 (Arganza and Hoyo de los Lobos fault-line valleys, respectively), which are two straight valleys on a fault line, disrupting

the sinuosity of the park's fluvial network. Particularly spectacular is the linearity of the Hoyo de los Lobos valley, a 560-m long section of the Lobos River canyon, where fluvial deposits and outcrops of the substratum can be observed in the riverbed.

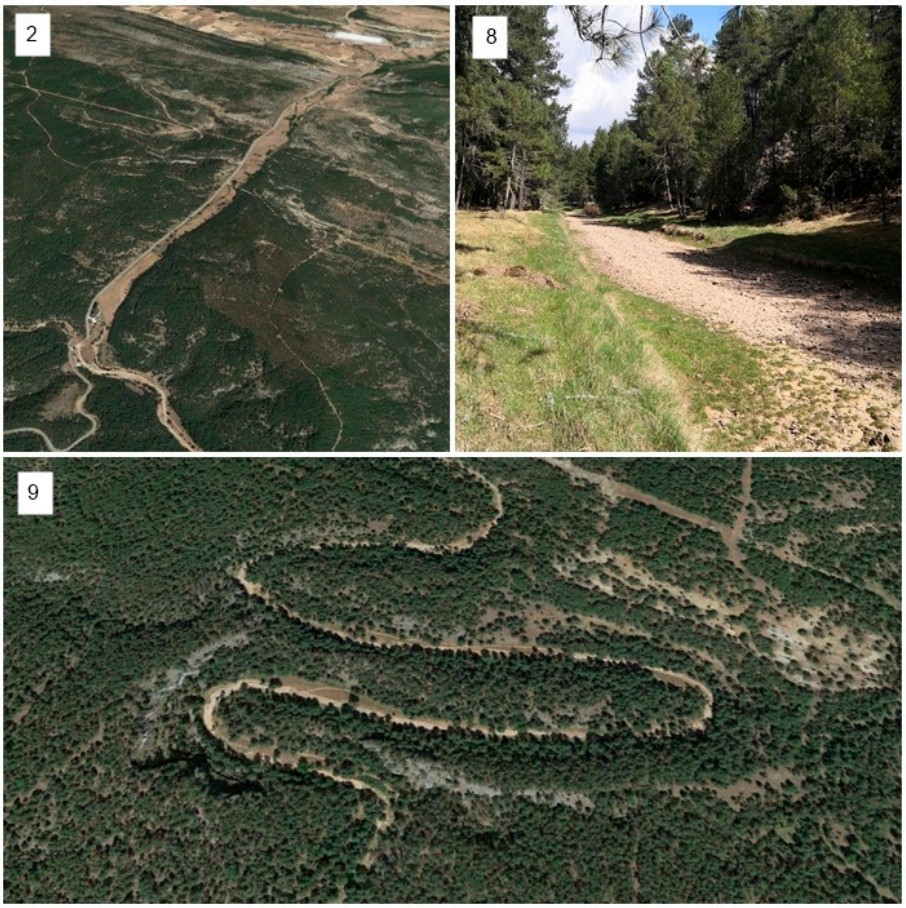

**Figure 5.** Fluvial attribution geomorphosites (N° 2, 8 and 9).

The Pico Navas slopeslide (N° 5), located in the northwestern corner of the park, is the only geomorphosite attributed to slope dynamics, standing out due to its current dynamics, big size and landscape content (Figure 6). It is an active rotational landslide formed by a body of blocks and a head with cracks, escarpments and uneven blocks easily visible.

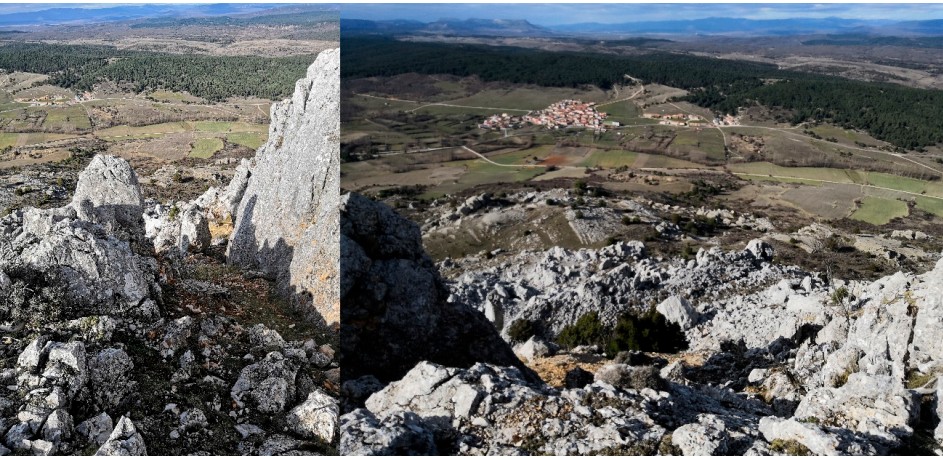

**Figure 6.** Slope attribution geomorphosite (N° 5).

To assess the interest of the fourteen geomorphosites, their representativeness, visibility and diversity of landforms were considered. Ten geomorphosites (71%) are of high interest, while four are of medium interest (29%). The high interest is generally due to the spectacular nature of landforms, the activity, the combination of landforms and, in some cases, their high landscape and cultural content. Those of medium interest have a low geomorphological interest or represent very common karstic landforms.

Accessibility is a crucial factor for the potential use of a geomorphosite, so those with difficult access or remoteness were initially discarded. Of the fourteen geomorphosites, seven have high accessibility, six have medium accessibility, and one, although it has low accessibility, has been maintained due to its high interest. Accessibility was assessed based on communications and infrastructure (paths, trails and tracks), as well as their state of conservation and the time required to reach the geomorphosite (distance and slope). The geomorphosites of Río Lobos–San Bartolomé Canyon, Ucero spring and the Galiana karstic system are highly accessible because they are very popular tourist sites. The remaining ones are hardly known and are not frequently visited, but they are easily accessible via hiking routes.

In summary, most of the geomorphosites are located in low-traffic areas with moderate impacts, mainly related to the visual impact of tracks and trails, as well as livestock and forestry activities. The overall conservation status is good, as the RLNP has not experienced a surge in nature tourism over the past decade. The majority of tourist activity is concentrated in a few specific locations, such as Ucero, La Galiana, Río Lobos–San Bartolomé Canyon and Siete Ojos Bridge, where the impact is more significant. These areas exhibit signs of increased human activity, including tracks, altered riverbanks, parking areas, urbanization that alters natural landscapes, erosion on slopes due to frequentation and damage to cavities with graffiti, breakage and footprints. Consequently, the conservation level in these sites has decreased.

### 4.2. Geotourist Assessment of Geomorphosites

Río Lobos geomorphosites are attractive territorial resources for visitors to the park. They are not only valuable for their geomorphological features but also for their landscape, aesthetic, cultural and educational significance. It is clear that they have tourism potential, especially for those visitors who are interested in acquiring knowledge and understanding the landscape, its dynamics and the complex relationships of natural elements, rather than just recreational or aesthetic enjoyment of natural heritage.

The results of applying the tourism potential assessment method (described in Section 3) indicate that, out of the fourteen geomorphosites inventoried in the park, four (28%) have high tourism potential, six (44%) have medium potential and four (28%) have low potential (Table 4).

Geomorphosites with high tourist potential are characterized by their scenic and cultural values, good conservation status and good accessibility. The most valuable geomorphosite is Río Lobos–San Bartolomé Canyon (N°1), where the scenic and cultural content generates a high-value space, being the most attractive but also the most frequented due to its multiple attractions, accessibility and proximity to parking. The Costalago orthocline valley (geomorphosite N° 10) is another highly attractive site, with a combination of elements, such as a viewpoint, valley, equipment and a high landscape value, making it a great tourist attraction. Additionally, geomorphosites N° 4, the Virgen de la Cueva synclinal flank, and N° 6, the Pico Navas perched syncline, combine cultural and aesthetic elements in unique environments, giving them a high tourism potential. These sites are valuable territorial resources with high geomorphological and didactic value, as well as significant geotourist potential.

**Table 4.** Geomorphosite tourism potential assessment in Río Lobos Natural Park.

| N° | Geomorphosites | Scenic | Scientific | Cultural | Educational | Conservation | | | Added | | | Geotouristic Potential |
| | | | | | | Vulnerability | Use Constraints | Accessibility | Security | Observation | Infrastructure, Goods | |
|---|---|---|---|---|---|---|---|---|---|---|---|---|
| 1 | Río Lobos–San Bartolomé Canyon | 15 | 5 | 10 | 15 | 5 | 5 | 10 | 10 | 15 | 5 | 95 |
| 2 | Arganza fault-line valley | 5 | 10 | 10 | 0 | 10 | 5 | 10 | 5 | 5 | 5 | 60 |
| 3 | La Sierra syncline flank crest | 15 | 10 | 10 | 5 | 10 | 5 | 5 | 5 | 15 | 0 | 75 |
| 4 | Virgen de la Cueva syncline flank | 15 | 15 | 5 | 15 | 10 | 5 | 10 | 10 | 15 | 5 | 95 |
| 5 | Pico Navas slopeslide | 5 | 5 | 5 | 0 | 10 | 5 | 0 | 5 | 10 | 0 | 45 |
| 6 | Pico Navas perched syncline | 15 | 15 | 10 | 15 | 10 | 5 | 5 | 5 | 15 | 0 | 90 |
| 7 | Las Raideras river sink | 10 | 10 | 10 | 0 | 10 | 5 | 10 | 10 | 10 | 5 | 75 |
| 8 | Hoyo de los Lobos syncline flank | 10 | 10 | 5 | 0 | 10 | 5 | 5 | 10 | 5 | 0 | 55 |
| 9 | La Isla entrenched meander | 15 | 15 | 5 | 0 | 10 | 5 | 5 | 10 | 15 | 0 | 75 |
| 10 | Costalago orthocline valley | 15 | 15 | 10 | 15 | 10 | 5 | 5 | 10 | 15 | 5 | 95 |
| 11 | Las Tainas y el Torcajón pits and karstic área | 10 | 10 | 5 | 10 | 10 | 5 | 5 | 5 | 5 | 0 | 60 |
| 12 | La Galiana karstic system | 15 | 15 | 10 | 10 | 5/10 | 0/5 | 5/10 | 5/10 | 15 | 5 | 75/85 |
| 13 | Ucero springs | 5 | 5 | 5 | 5 | 5 | 5 | 10 | 10 | 15 | 5 | 70 |
| 14 | Chorrón fluvial sink | 10 | 0 | 10 | 5 | 10 | 5 | 5 | 10 | 5 | 0 | 70 |

The majority of the geomorphosites (44%) are considered to have medium value, defined by a combination of aesthetic beauty and scenic interest. Some of these sites have lower value due to their specialization (N° 7, N° 13 and N° 14), as well as their remoteness and poor accessibility. Additionally, four geomorphosites (28%) are classified as having low value, primarily due to poor accessibility, moderate scenic values or their specialization, which detract from their attractiveness to tourists. These sites also have low educational value, despite being in a good state of conservation.

### 4.3. Geotouristic Map and Routes

The objective of geotouristic maps is to provide educational materials that explain the abiotic features of the area, with the aim of enhancing the level of recreation, culture and education in outdoor activities within natural protected areas. Based on the Río Lobos Natural Park inventory and geomorphosite tourism assessment, a geotouristic map was created. This map includes information about visible geomorphological elements and provides details for hikers such as trails linking geomorphosites, shelters, springs and elements of natural and cultural significance. The resulting map serves as a valuable tool for promoting geotourism and providing support in the field for local guides, monitors and tourist hikers who want to interpret nature and understand the landscape through direct knowledge of the terrain (Figure 7).

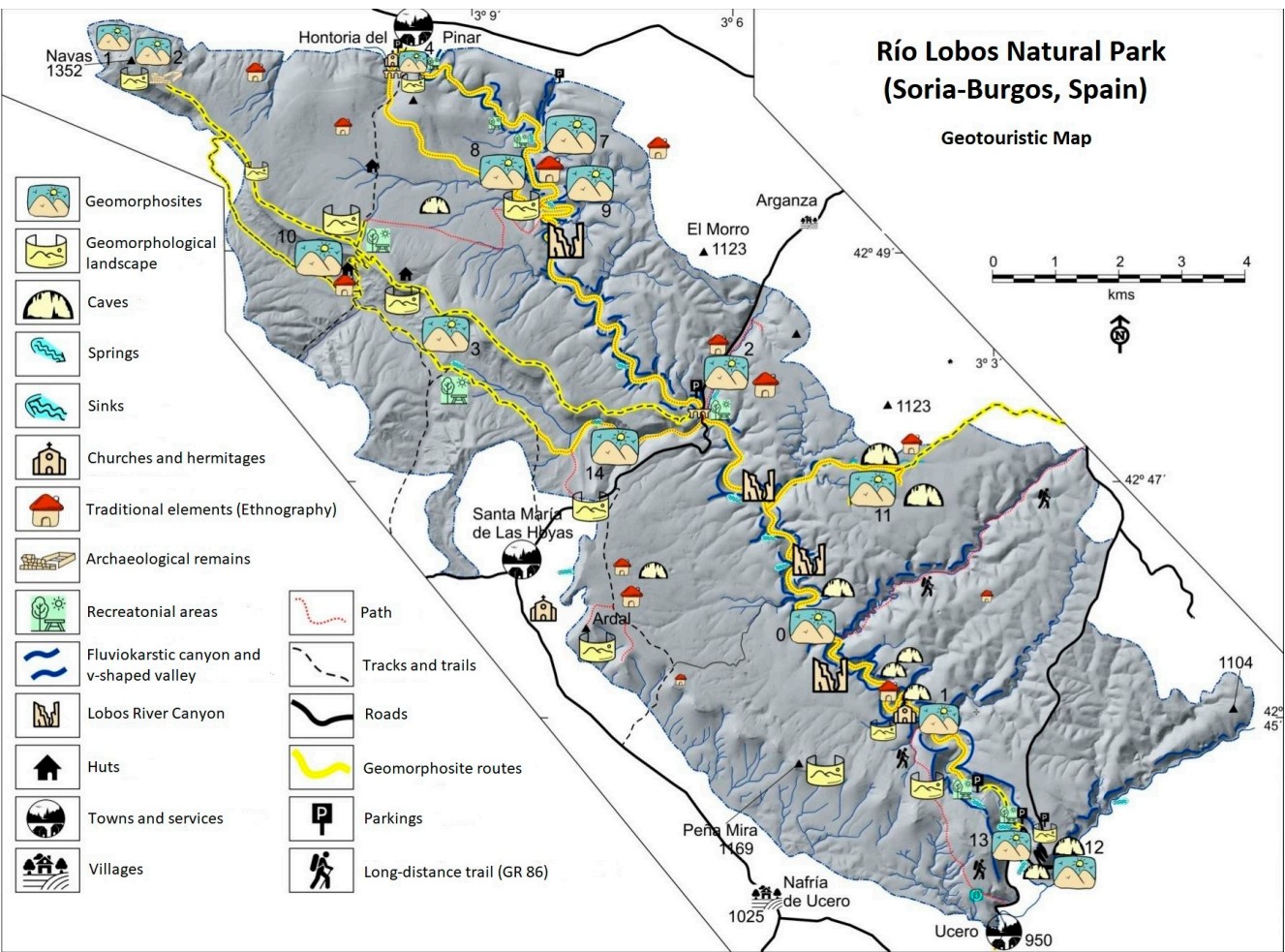

**Figure 7.** Geotouristic map in the Cañón de Río Lobos Natural Park. Numbers close to the Geomorphosite symbol indicate the Geomorphosite number in Tables 3 and 4.

The proposed map represents the park's topography, geomorphological characteristics and cultural heritage. It includes five reading levels: planimetry, altimetry, geomorphology,

human uses and tourist routes. Five routes are indicated, which encompass several of the geomorphosites inventoried and allow a variety of geomorphological landscapes to be explored. The five routes run along marked trails and tracks in the park, ensuring accessibility. Each route is described in terms of type (traverse, circular route), distance and slope, difficulty, accessibility, the geomorphosites that are traversed, cultural elements present along the route, itinerary and additional information (Tables 5–9). Detailed maps for each route have also been prepared (Figure 8).

**Table 5.** Route 1: The fluviokarstic canyon and karstic platform.

| Route | Trek |
|---|---|
| Distance/slope | 14 km/170 m |
| Difficulty | Gentle, long hike |
| Accessibility | High, marked trails, GR and PR |
| Geomorphosites | 13. Ucero Springs: representative element, karstic |
| | 1. Río Lobos–San Bartolomé Canyon: representative place, fluviokarstic |
| | 11. Las Tainas y el Torcajón pits and karstic area: representative place, karstic |
| Culture | Hermitage, Cueva Grande, apiary, Cañada real, tainas |
| Itinerary | El Congosto, Zabarrascal path, headwaters of the Caño, Las Tainas, El Torcajón, ravine of Las Fuentes, Pozo Perín, El apretado, Cueva Negra, Valderrueda (GR-86), Colmenar de los Frailes, Ermita de san Bartolomé, Valdecea. |
| Information | Signposted route, hiking guides |

**Table 6.** Route 2: Eastern valleys and mountain ranges: from Costalago to Chorrón through valleys and mountain ranges.

| Route | Trek |
|---|---|
| Distance/slope | 15.8 km/245 m |
| Difficulty | High, long traverse through the mountains |
| Accessibility | Good, marked trails and paths |
| Geomorphosites | 3. La Sierra syncline flank crest: representative place, structural |
| | 10. Costalago orthocline valley: exceptional place, structural |
| | 14. Chorrón fluvial sink: representative element, karstic |
| Culture | Majadas, legends and history, livestock uses, pasture, Pino spring, toponymy (Costalago, centenal, spring, La Raya, Chorrón, Raso pelado, La Gayuba), sheepfolds and enclosures, watchtower |
| Itinerary | Costalago Valley, dehesa de Santa María de las Hoyas, Hoz valley, Chorrón sink, Hoz ravine, Siete Ojos Bridge, Raso Pelado, La Sierra, La Gayuba head, Costalago viewpoint, El Portillo, Costalago valley |
| Information | Partially signposted route |

**Table 7.** Route 3: Río Lobos Natural Park peaks: Pico Navas and La Sierra.

| Route | Circular |
|---|---|
| Distance/slope | 14.5 km/282 m |
| Difficulty | High, route along stretches of trails and karren. If we discard the Portillo Ancho and the return by the pass of La Sierra, we avoid the trails and the access is by tracks |
| Accessibility | Medium, pine forest trails in some stretches, last stretch on karren |
| Geomorphosites | 10. Costalago orthocline valley: exceptional place, structural |
| | 3. La Sierra syncline flank crest: representative place, structural |
| | 5. Pico Navas slopeslide: exceptional element, slopes |

**Table 8.** Route 4: Burgos province canyons and platforms. Hontoria–Río Lobos–Hocinos–Hontoria.

| Route | Circular |
|---|---|
| Distance/slope | 10 km/200 m |
| Difficulty | Low, moderate-slope trails |
| Accessibility | High, well-marked trails |
| Geomorphosites | 7. Las Raideras river sink: representative element, karstic |
| | 8. Hoyo de los Lobos syncline flank: exceptional place, structural |
| | 9. La Isla entrenched valley: exceptional place, structural |
| | 4. Virgen de la Cueva syncline flank: representative place, structural |
| Culture | Hut of resineros, Roman road, Roman bridge, Romanesque hermitage (XI century) |
| Itinerary | Hontoria del Pinar, El castro, Agualino, el Apretadero, Chozo, Hoyo de Lobos, La Isla, Tres vallejos, El Hocino, Roman bridge, La Cueva Hermitage, Hontoria del Pinar. |
| Information | Signposted route, hiking guides |

**Table 9.** Route 5: A walk through the karst: viewpoints and caves.

| Route | Circular Walk with Access to the Caves |
|---|---|
| Distance/slope | 0.7 km/60 m |
| Difficulty | Moderate. walk through the caves, viewpoint and platform |
| Accessibility | High, parking, trails and viewpoints. If the caves are accessed, the rating changes and must be done with a guide |
| Geomorphosites | 12. La Galiana karstic system: representative element, karstic |
| | 13. Ucero springs: representative element, karstic |
| Culture | Cultural landscape |
| Itinerary | Parking, La Galiana viewpoint, La Galiana Alta, Sima La Galiana, parking |
| Information | Not signposted route, signs in the caves. The route through the caves must be guided |

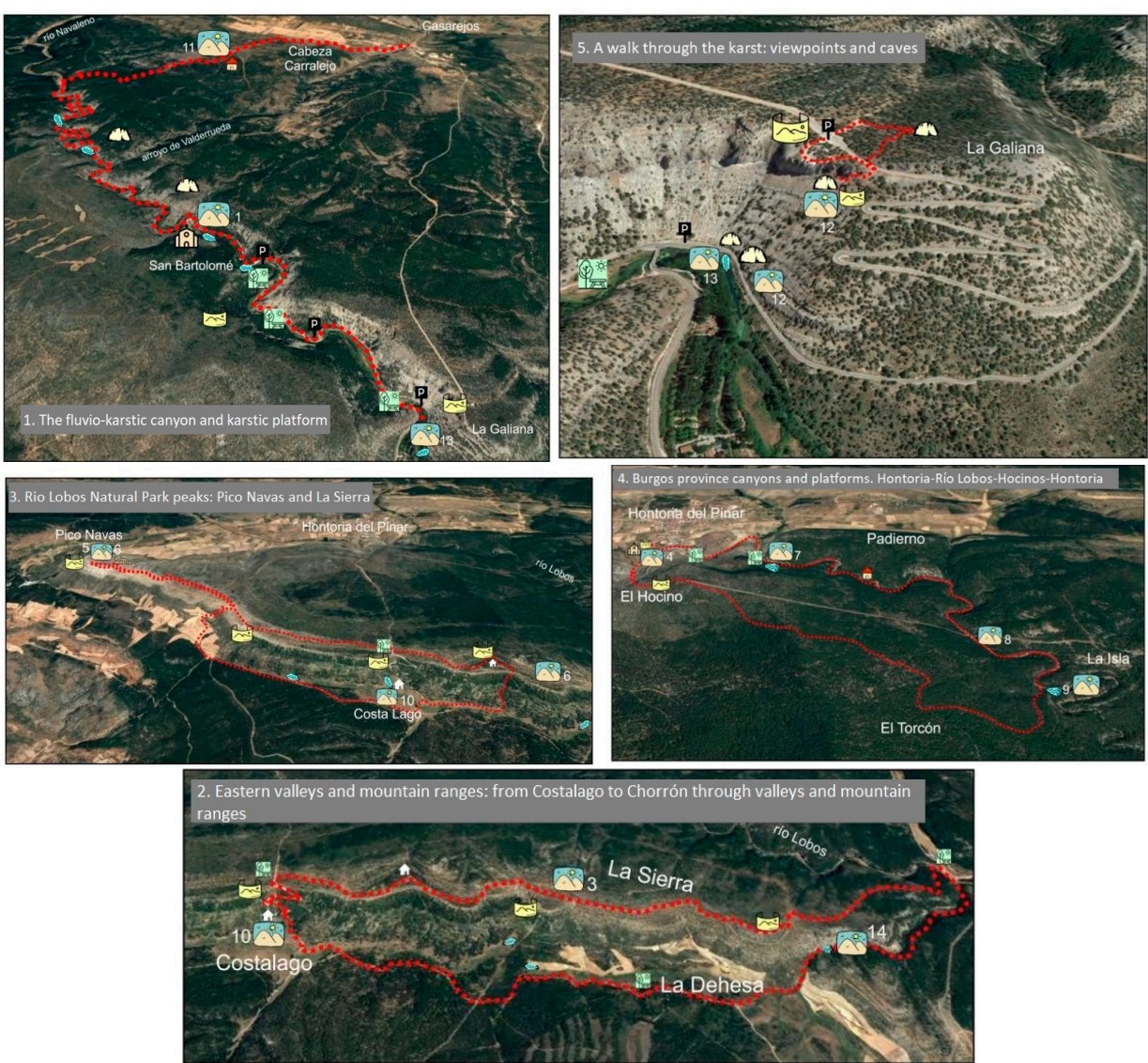

**Figure 8.** Detailed geotourist routes maps.

The routes are designed for both tourists and hikers, as well as for educational trips at various levels. They are centred around the concept of walking, learning about natural heritage elements along the way, and enjoying direct contact with nature and landscapes. The aim of these itineraries is, therefore, to combine leisure and education of tourists and students, integrating the interpretation of abiotic elements and landscapes with the experience of walking through the RLNP, from one geomorphosite to another. These geotourism routes serve as integrated tools for action, reflection and feeling in an exceptional natural environment, such as the Río Lobos Natural Park.

## 5. Discussion

In Spain, particularly in the Castilla y León region, there is a significant lack of geomorphosite inventory in NPAs. Geomorphology and abiotic elements of the environment are undervalued, despite being the basis of the landscape and supporting biodiversity, human and land uses. Visitors receive more information on flora, fauna and cultural heritage, while information on geomorphology and abiotic natural heritage is scarce or non-existent. There is a consensus among researchers on the importance of recognizing and promoting these sites in NPAs [32–35].

There is a lack of effective collaboration between researchers and NPAs (interpreters, public administrations, interpretation centres, etc.). The results of geomorphosite inventory and evaluation by research groups do not receive support beyond the university environment, leading to a lack of dissemination of this knowledge among the public visiting the NPAs and the general population. The managers of NPAs do not incorporate these studies into the values of these areas; therefore, it is essential to improve communication between researchers, managers and the general public. This can be achieved through geoheritage dissemination projects and activities in which universities involve the local population in their research, as has been achieved by research groups in other universities [36,37].

In areas such as the Río Lobos Natural Park, the geotourism offerings are limited to well-known and frequently visited sites, leading to overcrowding at the most accessible and diverse geomorphosites, such as the canyon and the San Bartolomé hermitage. There is a lack of commitment to diversifying the routes offered and promoting lesser known but equally interesting geomorphosites from a natural, geomorphological and scenic point of view.

Diversifying the routes could lead to a change in the tourism model in Río Lobos Canyon. Instead of predominantly short-term visitors, who only visit the canyon and the hermitage, without staying overnight, a multi-day tourism model could be promoted, offering a wider range of places to visit and encouraging visitors to stay longer and explore the park from a broader perspective, encompassing natural—abiotic and biotic—and cultural heritages. This could have a positive impact on the economy of the municipalities in and around the park, fostering local development initiatives through the promotion of its natural heritage.

## 6. Conclusions

The Río Lobos Natural Park is a valuable natural protected area that boasts a rich combination of natural and cultural heritage. The park's geomorphological features, well represented by the fourteen inventoried geomorphosites, are essential for scientific, educational and tourism purposes, providing insight into the landscape and land use. This geoheritage has not been previously studied or emphasized, making the results in this article a significant contribution to the scientific knowledge and dissemination of the park's values. Furthermore, the methodology used has proven effective in integrating geomorphosites into the management of the RLNP as geotourism resources and in the territorial use of geomorphological elements.

The tourism potential assessment methodology applied has identified ten geomorphosites with high and medium tourism potential (72%) and four with a low value (28%). The presented geotouristic map displays the geomorphosites' location, distribution and connection, serving as a useful tool for park's managers, interpreters and visitors, providing a clear overview of the distribution of the geomorphosites and the park's tourism and recreational offerings.

The five georoutes cover all geomorphosites and the major cultural sites, expressed in a straightforward manner on the geotouristic map. Georoutes serve as a tourist resource that combines leisure and education, showcasing the park's natural and cultural heritage and diversifying the tourist experience while highlighting lesser known but scientifically, pedagogically and geotouristically significant geomorphosites.

**Author Contributions:** Conceptualization, R.M.R.-P., M.J.G.-A. and E.S.; methodology, R.M.R.-P., M.J.G.-A. and E.S.; investigation, R.M.R.-P., M.J.G.-A. and E.S.; writing—original draft preparation, R.M.R.-P.; writing—review and editing, R.M.R.-P., M.J.G.-A. and E.S. All authors have read and agreed to the published version of the manuscript.

**Funding:** This research was funded by Junta de Castilla y León, grant number VA029G18. R.M.R.-P. holds a pre-doctoral fellowship founded by University of Valladolid.

**Data Availability Statement:** No new data were created or analyzed in this study. Data sharing is not applicable to this article.

**Conflicts of Interest:** The authors declare no conflicts of interest.

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
