# Peer review of "Geomorphosites as Geotouristic Resources: Assessment of Geomorphological Heritage for Local Development in the Río Lobos Natural Park"

_land, doi:10.3390/land13020128_

Round 1

Reviewer 1 Report

Comments and Suggestions for Authors

General comments

The paper discusses the inventory and assessment of geomorphosites within a Spanish NPA. While the topic aligns with the journal's scope, the manuscript lacks logical coherence, particularly in its structure and methodology section. Therefore, the paper in its current form cannot be published and should be re-worked. Here are some comments and recommendations are made in order to improve the final version.

To address these issues, consider the following suggestions:

-  Introduction: The current introduction is overly general and lacks a clear focus on the assessment of tourism potential in geomorphosites. To improve this section:

1. Focus specifically on the tourism potential within geomorphosites and why this assessment is critical for the Rio Lobos NPA in Spain.

2. Provide context by highlighting the relevance of assessing tourism potential in geomorphosites, especially concerning visitors, conservation efforts, and NPA management.

3. Establish a clear link between the introduction and the methodology section by illustrating how identified issues or objectives influenced the chosen methodology for assessing tourism potential.

- Methodology Section: This section needs significant restructuring and should focus solely on methodological aspects. I recommend dividing it into three sections:

2.1. Inventory of Geomorphosites: describe the process and criteria used for identifying and cataloging geomorphosites within the Rio Lobos NPA.

2.2. Qualitative Assessment of Geomorphosites' Tourism Potential: Provide a brief literature review, outline the methodology used for assessing tourism potential, and detail specific criteria and metrics employed to evaluate the attractiveness and suitability of these sites for tourism purposes.

2.3 Development of Geotourism Map: explain how the geotourism map was created based on the inventory and assessment of geomorphosites. Discuss the visualization techniques, cartographic elements, and information included in the map to effectively represent the tourism potential of these sites. Literature on geomorphosites mapping is insufficiently quoted for an international journal. ensure the paper includes sufficient references to literature on geomorphosites mapping for credibility in an international journal. Some papers have been recently published on this topic.

By restructuring along these lines, your paper will present a clearer progression of the study. Ensure each section is comprehensive and logically connected to the main objective of assessing geomorphosites for tourism potential within the Rio Lobos NPA. Using a workflow figure can aid readers in understanding your research.

Pay attention to the use of the terms 'values’ and ‘criteria’ (section 2), which are often not used correctly or as they are typically employed in international scientific literature. Review the usage of these terms in your text carefully, and if necessary, clarify their definitions or provide specific contexts to avoid misunderstandings or ambiguities.

Specific comments: see annotated manuscript.

Author Response

Dear reviewer: We appreciate your comments and suggestions for this article. 

Regarding the introduction, we have modified it to focus more on geotourism, and we have improved the context and the most relevant bibliographical references.

As you suggested, we have structured the methodology in three sections, to make it easier to understand the different steps. We consider that there is no place in this article for an exhaustive description of the geomorphosite inventory method, since, as we explained in line 87, it is not new and has been explained and applied in numerous articles, such as those cited in the same line. In the same way, we believe that the bibliographical review of the most relevant geotourism potential assessment methods is summarised in table 1, in addition to citing other authors, with up to five methodological references. As well as the criteria and metrics, which have been detailed in table 2. As for the third section of the methodology, the geotourism map, we do not focus on the mapping of geomorphosites as you pointed out, but on the elaboration of geotourism maps, having already cited the references on this subject that we have followed.

We have also modified the terms "value" and "criteria" to adapt them to the international expression.

Thank you for your comments.

Reviewer 2 Report

Comments and Suggestions for Authors

The aims of paper of the introduction should be written more clearly and more differentiated within the test. (pp.2).

Because of the tourist valuation approach presented in this paper, I believe that more information should be given about the San Bartolome Hermitage (such as the presence of sculpture motifs with a Templar theme) and its historical significance.

It would be of interest to include an assessment of the susceptibility and risk of natural and anthropic degradation of the points included in the paper, thus improving the assessment of their vulnerability.

Author Response

Dear reviewer: thank you very much for your comments and suggestions to improve the article. 

As you suggested, we have separated the objectives of the article into a separate paragraph in the introduction, so that it is distinct from the text. 

We have also added information about the hermitage of San Bartolomé in line 177. 

It would certainly be of interest to include an assessment of the susceptibility to degradation, although it is not our aim in this paper, we have not studied that aspect in depth so the focus of the article in that case would be different.

Thank you for your suggestions. Best regards

Reviewer 3 Report

Comments and Suggestions for Authors

Dear Author(s),

Thank you for the opportunity to read the paper entitled  GEOMORPHOSITES AS GEOTOURISTIC RESOURCES. ASSESSMENT OF GEOMORPHOLOGICAL HERITAGE FOR LOCAL DEVELOPMENT IN THE RÍO LOBOS NATURAL PARK.

The study aimed to investigate geomorphosites in Rio Lobos Natural Park as geotourists resources for local development. This topic has not been studied enough, so this study is more than necessary and confirms the interest and originality of this research. It fills the literature gap. It points out the value of geomorphological sites and heritage for local development and tourism.

The manuscript is well-written, understandable, and easy to comprehend. It is clear and easy to read, even for someone who does not specialize in the subject.  The results are nicely presented through tables and figures. The quality of the presentation is very high.  

All cited references are relevant to the research and up-to-date. However, there are many references missing in the introduction part and case study area, and this has to be corrected. The introduction section is missing similar studies from Europe and the world. This is needed in order to have a discussion and expand it.

The topic of this paper is interesting, but certain improvements would be appreciated.

Abstract

Comment 1

The abstract gives the impression of an interesting article; it's a really nice opening. Great job! However, citations are not needed in the abstract. You should have removed them.

Keywords:

Comment 2

You have just three keywords. I recommend adding two more to increase the visibility of the article and citations.

For example nature-protected areas, tourism potentials, etc.

Introduction

Comment 3

Lines 29-30 - n has grown exponentially worldwide. References are missing here.

Comment 4

Lines 40-52 – Only one reference. Please, you need to cite it properly.

Comment 5

Lines 53-60 – No citations at all.

Comment 6

The introduction needs to present the background of the studies in a similar context. The research gap needs to be highlighted using examples that are context-based.

You should point out the contribution, novelty and originality of the study. This can be added before Line 69.

Methodology

Comment 7

Lines 81-84 This sentence is a little bit hard to understand. Can you modify it? Make it more comprehensible.

Comment 8

The methodology is clear even to someone who is not in this field. Just as it should be, it is excellently explained and described. 

3. Study area: Rio Lobos Natural Park

Comment 9

Lines 134–142: Citations?

This whole section does not have proper citations. Please add them.

It is very nicely explained and written, but there are no references. This is a scientific article, and citations are necessary.

4. Results: geomorphosites as a geotouristic resource in Río Lobos Natural Park, Spain

Comment 10

The results are nicely presented through tables and figures. Great job! Everything is very clear!

Discussion 

Comment 11

You should expand the discussion. Compare to the situation in other countries and to previous research. 

Comment 12

Line 486 “There is no effective collaboration between researchers and NPAs (interpreters, public administrations, interpretation centers, etc.).” How would you handle this? Improve it? What does previous research suggest? 

Conclusions 

Comment 13

The conclusion needs rework. What are the theoretical contributions and practical implications of your study? What are the limitations? Future research? These segments are missing. Point out the originality and novelty of your study.

Comment 14

Once again, thank you very much for the opportunity to read this interesting article. The article has serious potential, and it deals with a very popular topic in research, but certain improvements are very much needed. I look forward to reading your article again.

I wish you all the best!

Sincerely,

Reviewer 

Author Response

Dear reviewer: 

thank you very much for your comments and suggestions to improve the manuscript.

Following your suggestions, we have removed the citations from the Abstract and added two new keywords.  We have also modified the introduction to add new references and improve the background of similar studies as you suggested.

We have modified the sentence in lines 81-84, as it was not very well expressed.

Regarding the absence of citations in section 3, studies on the Rio Lobos Natural Park are very scarce or non-existent, so there is no literature on the subject that we can cite. The information presented is the result of our own research. 

We have also modified the discussion and conclusion sections, adding new citations and highlighting the contributions and novelty of our work. 

Thank you for your comments. Best regards

Round 2

Reviewer 1 Report

Comments and Suggestions for Authors

I have carefully read the revised version and I thank the authors for incorporating my suggestions. The article now appears well-structured to me.

Author Response

Dear reviewer,

thank you for your comments to improve our manuscript.

Best regards